# Cardiac Arrhythmia Risk after Anti-Cancer Drug Exposure and Related Disease Molecular Imaging Outlook: A Systematic Review, Meta-Analysis, and Network Meta-Analysis

**DOI:** 10.3390/biology13070465

**Published:** 2024-06-25

**Authors:** Hongzheng Li, Wenwen Yang, Yuxuan Peng, Mingyan Huang, Feifei Liao, Aimei Lu, Zikai Yu, Xin Zhao

**Affiliations:** 1Postdoctoral Management Office, China Academy of Chinese Medical Sciences, Dongcheng District, Beijing 100700, China; 20180931809@bucm.edu.cn; 2Graduate School, China Academy of Chinese Medical Sciences, Dongcheng District, Beijing 100700, China; werweny@163.com (W.Y.); huangmingyan011@163.com (M.H.); 3Graduate School, Beijing University of Chinese Medicine, Chaoyang District, Beijing 100029, China; 20210931914@bucm.edu.cn (Y.P.); 20220931933@bucm.edu.cn (F.L.); 20210931915@bucm.edu.cn (A.L.)

**Keywords:** anthracyclines, arrhythmia, meta-analysis, network meta-analysis

## Abstract

**Simple Summary:**

Anthracyclines have a central role in anti-cancer therapy, but their side effects of chemotherapy-induced cardiotoxicity (AIC) during systemic administration have always been a problem for both patients and clinicians. The application of anthracycline agents leads to arrhythmias, causing nonspecific ECG changes. And the incidence rates of abnormal QT changes could be early markers of AIC. Therefore, we conducted a systematic review, meta-analysis, and network meta-analysis to evaluate the arrhythmic risk of anthracyclines and the comparative risk for each agent. We included 4 cohort studies, 8 RCTs, and 18 single-arm studies, and the result shows anthracyclines were associated with a statistically significant 90% increase in the risk of arrhythmia and a 114% increase in the risk of supraventricular arrhythmia. Among them, epirubicin ranked the most likely to have the highest risk of arrhythmia compared with non-anthracycline antineoplastic drugs in the analysis. The purpose of this research is to improve the monitoring of ECGs by clinicians while applying anthracyclines and to try to use molecular imaging technology to study the mechanism of drug action on cardiac electrophysiology in the future. The effects of anthracycline agents on the electrophysiological properties of cardiac cells can be better understood with the help of imaging molecular targets in cardiac cells, which may result in safer and more effective clinical use of the drugs.

**Abstract:**

Background: Chemotherapy is the main first-line treatment, but there is a problem of adverse reactions to systemic drugs. Chemotherapeutic agents may cause adverse effects on the body, influencing the prognosis. Whether the clinical application of anthracyclines is associated with an increased arrhythmic risk remains controversial. To evaluate the arrhythmic risk of anthracyclines as a class, and the comparative risk for each drug, we conducted a systematic review, meta-analysis, and network meta-analysis. Methods: PubMed, Web of Science, EMBASE, and the Cochrane Library were searched, up to March 2022, for randomized controlled trials, cohort studies, and case–control studies that investigated the association between anthracyclines treatment and the risk of arrhythmia. We followed the PRISMA 2020 guidelines for data selection and extraction. Outcomes were pooled using fixed effects models in cohort studies and randomized controlled studies, and random models in single-arm studies. Direct and indirect comparisons in network meta-analysis were performed using frequentist methods. Results: In total, 4 cohort studies, 8 RCTs, and 18 single-arm studies were included in our analysis. Anthracyclines’ use was associated with a statistically significant 90% increase in the risk of arrhythmia (odds ratio [OR] 1.90; 95% confidence interval [CI] 1.62–2.24) and a 114% increase in the risk of supraventricular arrhythmia (OR 2.14; 95% CI 1.18–3.89). And the single-arm studies also indicated that the incidence of arrhythmia rate is 20% and the 95% CI is 15/100–25/100. Epirubicin ranked most likely to have the highest risk of arrhythmia compared with non-anthracycline antineoplastic drugs in the analysis (OR 43.07 [95% CI 2.80–2105.83]) by network meta-analysis. Conclusions: Our findings show a significant association between anthracyclines’ use and an increased risk of arrhythmia, especially supraventricular arrhythmia. Epirubicin ranked with the highest probability of arrhythmia. These results indicated that cardiac rhythm should be strictly monitored during the application of anthracyclines in clinical practice, and a possible therapy for anthracycline-associated arrhythmia should be explored. Molecular imaging technology is an important means to study the mechanism of drug action on cardiac electrophysiology in the future. By imaging molecular targets in cardiac cells, the effects of drugs on the electrophysiological properties of cardiac cells can be understood, which provides information for the development of safer and more effective drugs.

## 1. Introduction

Chemotherapy is the predominant first-line treatment of various cancers at present [1], but its side effects during systemic administration have always been a problem for patients and doctors. Among chemotherapeutic treatments, the use of therapeutic drugs often has adverse effects on the body and may even affect the prognosis. Anthracyclines have a central role in anti-cancer therapy, which is widely used to treat lymphoma, sarcoma, breast cancer, and pediatric leukemia [2]. However, their cardiac toxicity remains a serious clinical problem and has become the primary cause of chemotherapy-induced cardiotoxicity (AIC) [3]. The estimated prevalence rate for all kinds of AIC is 16–23% [4]. Previous studies have shown that patients given anthracycline agents are at risk of heart failure, cardiomyopathy, and even death, with a significantly impaired life expectancy and quality of life.

Anthracyclines are glycoside drugs comprising the amino sugar daunosamine linked to a hydroxy anthraquinone aglycone, and they act by DNA intercalation, oxidative stress generation, and topoisomerase II poisoning towards patients with cancers [5]. These pathological contributors lead to the development of cardiomyopathy including oxidative stress and intracellular iron, ultimately leading to apoptosis [6]. Importantly, these electrocardiograph (ECG) changes predate the decline in the ejection fraction in patients with obvious dose-dependent cardiac toxicity [7,8,9], which lead to a major cause of death in cancer survivors [10]. Recently, researchers have suggested that anthracycline agents may be a risk factor for arrhythmias, which cause nonspecific ECG changes, such as ST-T changes, QT interval prolongation, and atrial fibrillation (AF) [11,12,13,14,15]. Whereas, there have also been some studies demonstrating the importance of the left ventricular ejection fraction (LVEF) reduction or NT-BNP plasma level rather than the risk of arrhythmias [16,17]. Even though the observed arrhythmias were rather benign [18], the incidence rates of abnormally prolonged QT dispersion intervals, QT dispersion, and corrected QT dispersion could be early markers of AIC [19].

In addition, the existing evidence is relatively limited and inconclusive. This systematic review, meta-analysis, and network meta-analysis were conducted to comprehensively evaluate the risk of arrhythmias among patients given anthracycline agents compared to individuals without anthracycline treatment by identifying all relevant studies and combining their results.

## 2. Materials and Methods

### 2.1. Protocol

The protocol used for the current systematic review and network meta-analysis is included in the Appendix A. The protocol was registered in the PROSPERO registry (registration number CRD42021286232).

### 2.2. Study Design

A systematic review and network meta-analysis were performed with a frequentist statistical approach, based on a prespecified study protocol (Appendix A). The results of the present study are reported according to the PRISMA extension statement for network meta-analysis.

### 2.3. Search Strategy, Selection Criteria, and Data Extraction

We performed a systematic search of PubMed, EMBASE, Web of Science, and the Cochrane Library that investigated the association between anthracycline treatment and the risk of arrhythmia from inception to 18 March 2022; studies with missing data were excluded. The systematic search strategy and search terms are documented in the Appendix A.

We considered randomized controlled trials (RCTs), cohort studies, and case–control studies written in the English language and investigating the occurrence of arrhythmia of the main drug. Pharmacological agents are listed in Table 1. A manual search of the reference lists of review articles and original studies was performed to identify additional reports. No language was used during the search. Target studies were limited to populations with anthracycline-induced arrhythmia, enrolled in the outpatient setting or after stabilization following hospitalization. Articles were also excluded from the analysis if they had insufficient published data to estimate the risk ratio, odds ratio (OR), and confidence interval (CI). Studies were excluded when the entire population included patients with arrhythmia before adopting anthracyclines to reduce the potential for confounding by indication.

The studies were obtained from four medical databases and were independently screened by two reviewers (Hongzheng Li and Wenwen Yang) based on their titles and abstracts. A comprehensive search of the reference lists of the review articles and original studies was performed to identify additional reports. Disagreements were resolved through a consensus or referral to a third reviewer (Xin Zhao) when no consensus was reached. Data were extracted by two independent reviewers (Feifei Liao and Mingyan Huang). Disagreements were resolved through a consensus or referral to a third reviewer (Xin Zhao) when needed. Data regarding the following characteristics were extracted: study details (study design, geographical location, publication year, funding sources, and duration of follow-up); participants’ details (number, study population, age, and sex); intervention and comparator characteristics (drug name, dosage regimen); outcomes and measurements; and covariate adjustments. Data were collected independently by two reviewers (Zikai Yu and Mingyan Huang) and checked by a third reviewer (Yuxuan Peng).

### 2.4. Study Quality and Critical Appraisal

The quality of the studies was assessed according to the Newcastle–Ottawa Quality Assessment Scale Cohort Studies (NOS) and Risk of Bias-2 (RoB) for RCTs [20]. NOS includes three aspects (selection, comparability, and exposure for case–control studies or outcomes for cohort studies), as well as scores of 4, 2, and 3, respectively. These assessments of quality were undertaken by two reviewers (Hongzheng Li and Wenwen Yang). Any differences in the quality assessments were resolved by a consensus or referral to a third reviewer (Zikai Yu). The certainty of the evidence was evaluated by using the Grading of Recommendations Assessment, Development, and Evaluation (GRADE) approach that was specifically developed for concluding a network meta-analysis, and the GRADE approach adopted a minimally contextualized framework that has been described elsewhere in detail. The certainty of the evidence was expressed as high certainty and low certainty. The classification of intervention was expressed as Category 2 (among the most effective), Category 1 (inferior to the most effective, or superior to the least effective), and Category 0 (among the least effective). Trained GRADE methodologists analyzed the data to assess the quality of evidence, given the strength of the recommendation (Appendix A).

### 2.5. Outcomes

In network meta-analysis, we focused on any kind of arrhythmia as the primary outcome of interest, and we analyzed the outcome of supraventricular arrhythmia, tachycardia, and ventricular arrhythmias. Meanwhile, in pooled analysis in single-arm studies, we focused on the risk of arrhythmias.

### 2.6. Statistical Analysis

Raw data were extracted from individual studies using a predefined data extraction form created using Excel, and the pooled ORs and corresponding 95% CIs were calculated for each primary outcome. The heterogeneity of the data was quantified using the *Q* and *I*^2^ statistics. High heterogeneity was considered significant when *p* < 0.1 for the *Q* statistic or when *I*^2^ was >50%. The effect sizes of anthracyclines as a class were pooled using random-effects models. These analyses were performed using STATA software version 15.1. Prevalence was estimated based on the total population at risk and the number of anthracycline users by using the Poisson distribution for the calculation of 95% CIs, as appropriate, and STATA software version 15.1. The pooled prevalence was assessed using the Der Simonian and Laird random-effects model when significant heterogeneity was present (*I*^2^ > 50% or *p* < 0.005) or a fixed-effects model in the absence of significant heterogeneity.

Additionally, we conducted a network meta-analysis according to PRISMA-NMA 2020 to pool direct and indirect comparisons of the marketed anthracyclines (namely, epirubicin, doxorubicin, pegylated liposomal doxorubicin, and other anthracyclines) concerning their relative arrhythmia risks. This analysis and the network graphs were generated using the ‘GeMTC 0.14.3’ package in the R environment (version 4.1.3). Moreover, we ranked the risk of arrhythmia associated with anthracyclines using *p*-scores derived from network point estimates and standard errors. The ranking indicated the probability of being associated with the highest risk of arrhythmia, the second highest risk, the third highest risk, and so on. Rank 1 corresponded to the lowest and Rank N to the highest probability of inducing arrhythmia [21]. Inconsistency was assessed using the *Q* statistic and by comparing the results of direct and indirect estimates using the ‘GeMTC 0.14.3’ package in the R environment (version 4.1.3).

Possible publication bias was assessed using a graphical visualization of the funnel plot, Begg’s test, and Egger’s test [22]. A meta-regression was performed to identify influential variables; *p* < 0.05 indicated a significant effect. Sensitivity analyses were performed as post hoc analyses according to the study type and study period. We performed subgroup analyses according to the study design (cohort studies and RCTs). Because of the limited number of studies for each analysis, we did not perform a subgroup analysis of the included studies.

## 3. Results

### 3.1. Description of the Selected Studies

Our search generated 1236 records for evaluation. Duplicate records were removed before screening, resulting in 933 potentially relevant works that were further evaluated based on the abstract. After excluding irrelevant abstracts, 122 articles were selected for full-text evaluation. In total, 9 articles, including 3 cohort studies, 6 RCTs, and 18 single-arm studies, were included in our analysis. The search process is illustrated in Figure 1. The selected cohort studies and RCTs involved the use of anthracyclines as a group and the use of epirubicin, doxorubicin, pegylated liposomal doxorubicin, and rubidazone individually. The study characteristics (study design, geographical location, funding sources, duration of allocation, races, type of cancer, duration of treatment, duration of follow-up, sample size, mean age, gender, type of anthracyclines, treatment control, and quality scores, specifically) are summarized in Table 2. Additional information, including quality assessments of observational studies and the overall risk of bias of RCTs, is available in the Appendix A. The comparator group was most commonly treated with non-anthracycline antineoplastic drugs, such as cyclophosphamide, methotrexate, fluorouracil, carboplatin, and paclitaxel. The characteristics of the 18 single-arm studies are summarized in Table 3.

### 3.2. Bias Assessment

Among these 12 studies, 3 were industry-sponsored and 3 were sponsored by national science foundations. The three studies sponsored by the industry provided only 12.4% and 10.7% of the patients included in the meta-analysis and network meta-analysis, respectively. The study size was heterogeneous, ranging from 40 to 5026 patients in individual studies. The cancer types varied. Seven studies included patients with breast cancer. One study focused on advanced non-small-cell lung cancer. The remaining four studies included patients with non-Hodgkin’s lymphoma, ovarian cancer, prostate carcinoma, and various malignancies. There were some differences in the duration of follow-up of the studies. Four studies did not mention the follow-up duration and six studies involved more than 12 months of follow-up. However, one study had a follow-up duration of 6 months and one study had a follow-up duration of 6.5 months. Additionally, the anthracycline treatment varied. Five studies used epirubicin, two studies used doxorubicin, three studies used pegylated liposomal doxorubicin, and two studies used different types of anthracyclines. Although we assessed publication bias quantitatively, all studies exhibited a minimum partial response (Table 2, Appendix A).

### 3.3. Meta-Analysis

#### 3.3.1. Risk of Arrythmias

Nine studies with 8537 participants (three cohort studies and six RCTs) reported arrhythmia among anthracycline users. Regarding cohort studies, compared with non-anthracycline antineoplastic drug users, anthracycline users had a summary OR of 1.83 (95% CI, 1.56–2.16; three studies). Regarding RCTs, the summary OR was 5.54 (95% CI, 1.91–16.08; six studies) for anthracycline users. No significant differences were observed between subgroups (*I*^2^ = 0%, *p* = 0.431; *I*^2^ = 0%, *p* = 0.823). This pooled analysis showed a significant association between anthracycline use and an increased risk of arrhythmia (OR, 1.90; 95% CI, 1.62–2.24), with no heterogeneity (*p* = 0.580; *I*^2^ = 0%) (Figure 2). A funnel plot of the standard error of the logarithm of the transformed proportion is shown in Appendix A; Begg’s test and Egger’s test suggested no publication bias (*p* = 0.602; *t* = 7.53). 

#### 3.3.2. Risk of Supraventricular Arrhythmia

Three studies with 5543 participants (one cohort study and two RCTs) reported supraventricular arrhythmias among anthracycline users. Regarding cohort studies, compared with non-anthracycline antineoplastic drug users, anthracycline users had a summary OR of 1.85 (95% CI, 0.98–3.49; one study). Regarding RCTs, the summary OR was 6.34 (95% CI, 0.76–52.94; two studies) for anthracycline users. No significant differences were observed between subgroups (*I*^2^ = 0% and *p* = 0.839 for RCTs). The pooled analysis showed a significant association between anthracycline use and an increased risk of supraventricular arrhythmia (OR, 2.14; 95% CI, 1.18–3.89), with no heterogeneity (*p* = 0.543; *I*^2^ = 0%) (Figure 3). Begg’s test and Egger’s test suggested no publication bias (*p* = 1.0; *t* = 2.62).

#### 3.3.3. Risk of Arrythmia in Single-Arm Studies

The pooled analysis of the prevalence of anthracycline-associated arrhythmia included 6566 patients with 227 reported arrhythmias. The pooled prevalence of anthracycline users in our cohort was 20% (95% CI, 0.15–0.25), with significant heterogeneity between the studies (*I*^2^ = 93.2%; *p* = 0), as it shown in Figure 4. We repeated the analysis stratified by anthracycline type, geographical location, and cancer type. We found no regional variation and statistically similar prevalence estimates for North America (9 studies; 17%; 95% CI, 0.08–0.25), Europe (8 studies; 27%; 95% CI, 0.15–0.39), and Asia (1 study; 28%; 95% CI, 0.14–0.42). The prevalence was high in North America (*p* = 0; *I*^2^ = 92.7%) and Europe (*p* = 0; *I*^2^ = 93.2%). We repeated the analysis stratified by the decade of the study and anthracycline type; however, secular trends in the prevalence of arrhythmia in those studies were not observed.

### 3.4. Network Meta-Analysis

Four cohort studies and eight RCTs with 9875 patients were included in this network meta-analysis. Patients were treated with epirubicin plus non-anthracycline antineoplastic drugs in four studies, and with different anthracyclines in three studies. Among them, doxorubicin plus non-anthracycline antineoplastic drugs were used in two studies. Pegylated liposomal doxorubicin plus non-anthracycline antineoplastic drugs were used in one study. A comparison between anthracyclines and non-anthracycline antineoplastic drugs was performed in two studies. A comparison of different dosages of pegylated liposomal doxorubicin was performed in one study. A comparison of different dosages of epirubicin was performed in one study. A comparison of different dosages of pirarubicin was performed in one study. A comparison of different dosages of doxorubicin was performed in one study (Figure 5).

During the network meta-analysis of arrhythmia (network plot in Figure 6), epirubicin was ranked most likely to be associated with the highest risk of arrhythmia compared with non-anthracycline antineoplastic drugs (OR, 43.07; 95% CI, 2.80–2105.83), and non-anthracycline antineoplastic drugs were most likely to be associated with the lowest risk of arrhythmia (Figure 6). And a comparative ORs for the incidence of arrythmia with different anthracycline treatments and non-anthracycline treatments was showed in Figure 7. The probability of each intervention inducing the highest risk of arrhythmia was ranked according to seven possible positions (Table 4). Rank 7 corresponded to the highest probability of inducing arrhythmia and Rank 1 corresponded to the lowest probability of inducing arrhythmia. Besides, a ranking table is also provided (Appendix A). Doxorubicin and pegylated liposomal doxorubicin had the highest probability of inducing the highest risk of arrhythmia. Additionally, non-anthracycline antineoplastic drugs appeared to be associated with a lower risk of arrhythmia than other anthracycline treatments.

## 4. Discussion

This meta-analysis focused on the association between anthracyclines and the risk of arrhythmia. Our results identified a consistent association between anthracycline use and arrhythmia, with a statistically significant 90% increase in the risk of arrhythmia and 114% increase in the risk of supraventricular arrhythmia. According to network meta-analyses, epirubicin treatment was associated with the highest probability of being associated with the risk of arrhythmia compared to other anthracycline and non-anthracycline treatments. Overall, these findings support a link between arrhythmia and anthracyclines and provide a relatively rational explanation regarding why anthracyclines significantly increase the risks of cardiac events and mortality in addition to heart failure.

Anthracyclines are widely used to treat adult and pediatric cancers. Despite their therapeutic efficacy, anthracyclines are associated with both acute-onset and late-onset cardiac toxicities [50]. Meta-analyses have reported an overt cardiotoxicity incidence of 6.3% and a subclinical cardiotoxicity incidence of 17.9% [51]. Among these AICs, the induction of arrhythmia may not be as serious as cardiomyopathy or heart failure. However, anthracyclines are associated with electrocardiographic alterations [52]. In 1981, a patient with non-small-cell lung cancer developed acute-onset AF with a rapid ventricular response during the first administration of Adriamycin; thereafter, physicians became aware of possible life-threatening arrhythmias during or soon after Adriamycin administration [53]. However, anthracycline-induced arrhythmia did not gain much attention until Bender first described the prolongation of the QT interval with anthracycline therapy in 1984 [54]. Two years later, epirubicin was reported to cause transient cardiac arrhythmias and electrocardiogram alterations [55]. The impact of anti-cancer drugs on cardiac repolarization (i.e., QT prolongation) and the potential risk of life-threatening arrhythmia torsades de pointes, which are common challenges in the field, have been gradually noticed and reduced.

In our study, supraventricular arrhythmia was the most common adverse effect of anthracyclines. Previous studies have reported that AF, one of the representatives of supraventricular arrhythmia, was detected using Holter monitoring for 10.3% of patients treated with doxorubicin [56]. Studies have revealed that AF onset always occurs before the development of heart failure in patients and that the incidence of AF increases with anthracycline doses; therefore, new-onset AF might be a sensitive indicator of AICs [11,57]. An experimental study of sheep also reported that atrial remodeling favoring AF development was observed in doxorubicin-induced heart failure models [58]. Experiments involving rats also confirmed that anthracycline exposure resulted in an 86% incidence of ventricular tachycardia or ventricular fibrillation [59].

Although QT prolongation is not a perfect marker of the arrhythmia risk, it has become a primary safety metric for oncologists [60]. The variability in the QT interval, a probable marker of an arrhythmogenic substrate, reflects an increase in regional differences in ventricular repolarization [61] and has been associated with life-threatening ventricular arrhythmias and sudden death [62]. Our results also suggest that QT prolongation is common in anthracycline-induced arrhythmia, similar to the results of previous reports [63].

Left ventricular dysfunction caused by AIC has been believed to be irreversible; however, the early initiation of standard medical treatment for heart failure may lead to left ventricular functional recovery with AIC [64]. In this meta-analysis, we found that anthracycline use may increase arrhythmia-related risks. The close monitoring of ECG changes would clarify the occurrence of acute-onset and delayed-onset cardiac toxicity caused by anthracyclines, which may have some significance in preventing irreversible AIC and the occurrence of serious adverse cardiac events. Further studies of life-long cardiac follow-up for patients undergoing anthracycline-based chemotherapy are necessary [65].

Our study has several strengths. First, the results are generalizable because of the large number of participants involved in the arrhythmia analyses and the use of raw data. Subgroup, sensitive, and regression analyses were performed to support these findings. Furthermore, to our knowledge, this is the first study to conduct indirect comparisons using a network meta-analysis to assess differences in the arrhythmia risks associated with individual anthracyclines.

Our study has several potential limitations as well. First, our study was not designed to explore the interaction between comorbidity and the arrhythmic adverse effects of anthracyclines; therefore, the observed risk does not reflect the risk in special populations such as patients with conduction disorders and patients with prior cardiovascular events. Second, many of these studies did not provide information regarding important lifestyle factors that influence the arrhythmic risk, patient compliance, and chemotherapy regimen. Third, a misclassification bias may have occurred because arrhythmia screening is not sufficiently intensive to detect several paroxysmal arrhythmias such as paroxysmal AF and infrequent premature contractions. Fourth, the meta-analysis was highly heterogeneous among single-arm studies. The high heterogeneity can be attributed to different indications for the types of carcinomas and anthracyclines, different dosages of anthracyclines, different treatment and follow-up durations, different coexisting conditions, different countries, different ages, and different sexes. We used a random-effects model for our calculations to account for the possibility of study-dependent variations in effects and conducted a sensitivity analysis. Finally, this analysis was not sufficiently strong to provide the basis for any changes in practice. These findings need to be formally assessed by larger, prospective, real-world studies and clinical trials with strict ECG monitoring.

Molecular imaging techniques are increasingly being recognized as an important tool in the study of drug effects on cardiac electrophysiology. With advancements in science and technology, molecular imaging has become an essential approach to investigating the mechanism of drug action. By visualizing molecular targets within cardiac cells, we can gain valuable insights into how drugs impact the electrophysiological properties of the heart, thereby facilitating the development of safer and more effective medications. In the field of cardiac electrophysiology, molecular imaging can be employed to examine the influence of drugs on various molecular targets, including ion channels, receptors, and enzymes. For instance, anti-cancer drugs may alter the intracellular calcium ion channels in the heart, affecting the excitability and contractility of cardiac cells. Through molecular imaging, we can observe the effects of drugs on calcium channels, enabling a better understanding of their mechanisms and impacts. Furthermore, molecular imaging can also shed light on the effects of drugs on other molecular targets in cardiac cells, such as sodium channels, potassium channels, and hydrogen channels, all of which play crucial roles in the electrophysiological properties of the heart. Therefore, studying the effects of drugs on these channels using molecular imaging is vital for the development of safer and more effective medications. In conclusion, molecular imaging technology will undoubtedly emerge as a key approach in future studies on the mechanisms of drug action. By investigating the imaging of molecular targets within cardiac cells, we can enhance our understanding of how drugs affect the electrophysiological properties of the heart, thus providing essential information for the development of safer and more effective medications. At the same time, molecular imaging technology can also be used to study the mechanism of drug action in other organs and tissues, providing important reference information for drug development and clinical application.

## 5. Conclusions

The clinical application of anthracycline was associated with an increased risk of arrhythmia, especially supraventricular arrhythmia. Epirubicin ranked with the highest probability for arrhythmia. These results indicated that cardiac rhythm should be strictly monitored during the application of anthracyclines, and possible therapy for anthracyclines-associated arrhythmia should be explored.

## Figures and Tables

**Figure 1 biology-13-00465-f001:**
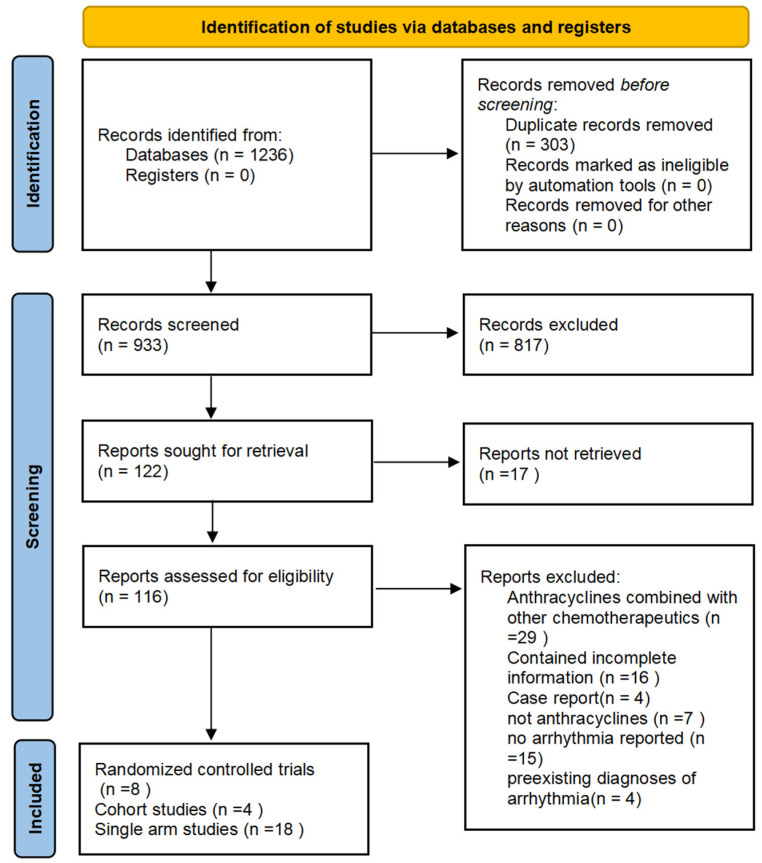
Selection process. Each article was retrieved, screened, and selected for the quantitative analysis.

**Figure 2 biology-13-00465-f002:**
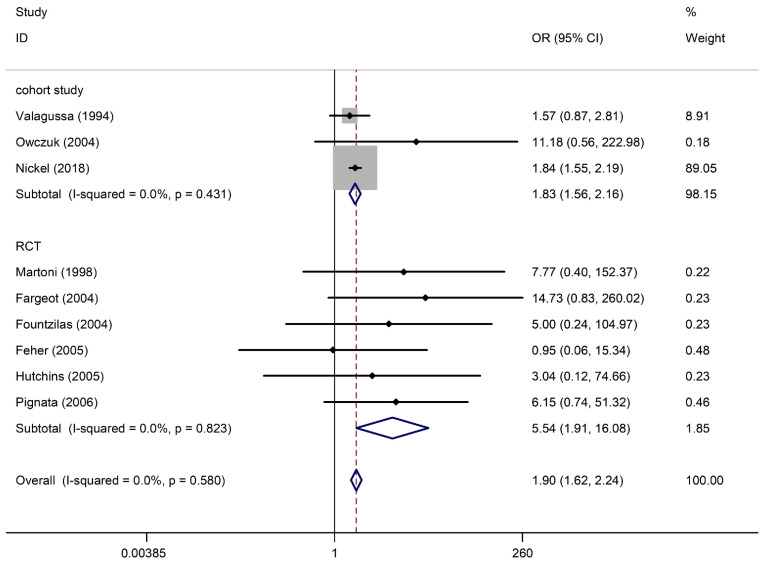
Arrhythmia in anthracycline users and nonusers. The forest plot demonstrates point estimates of the odds ratios (ORs) surrounded by 95% confidence intervals (CIs) calculated using the fixed-effects model [23,24,25,26,27,28,29,30,31]. RCT, randomized controlled trial.

**Figure 3 biology-13-00465-f003:**
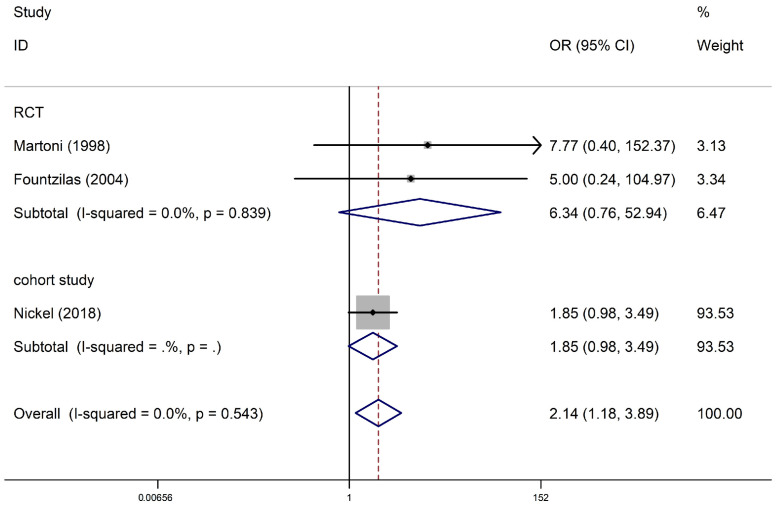
Supraventricular arrhythmia in anthracycline users and nonusers. The forest plot demonstrates point estimates of the odds ratios (ORs) surrounded by 95% confidence intervals (CIs) calculated using the fixed-effects model [24,27,31]. RCT, randomized controlled trial.

**Figure 4 biology-13-00465-f004:**
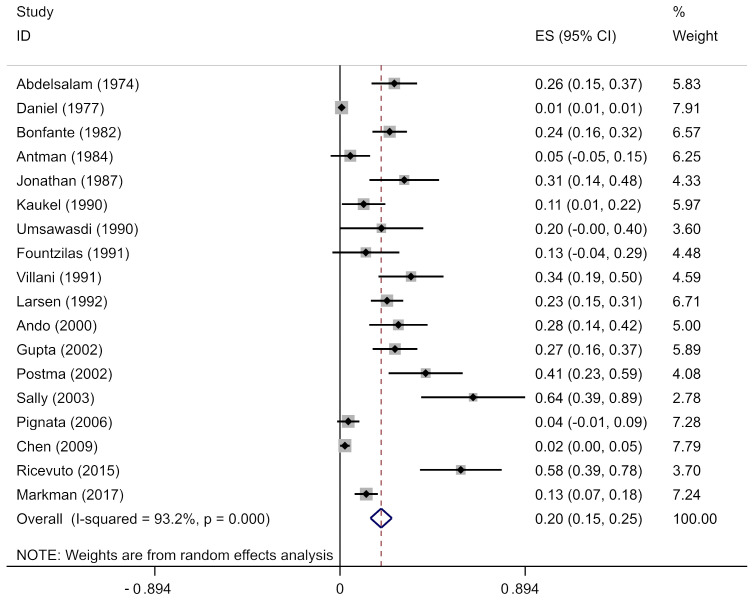
Arrhythmia in anthracycline users in single-arm studies. Prevalence estimates (ES) and confidence intervals (CIs) from studies with zero events were evaluated by adding 0.5 cases to both the numerator (number of anthracycline users) and the denominator (total number of arrhythmic events), consistent with the recommended practice [12,19,30,35,36,37,38,39,40,41,42,43,44,45,46,47,48,49].

**Figure 5 biology-13-00465-f005:**
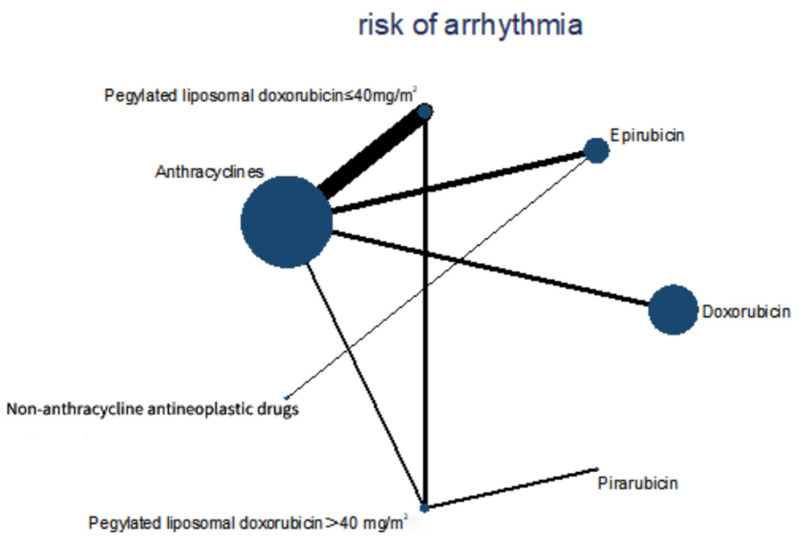
Comparisons during the Bayesian network meta-analysis. Lines connect the interventions that were studied during head-to-head (direct) comparisons of eligible controlled trials. The interrupted lines connect indirect comparisons. The size of the nodes is proportional to the number of patients (in parentheses) who received treatment. The width of the lines is proportional to the number of trials (next to the line) involving the connected treatment.

**Figure 6 biology-13-00465-f006:**
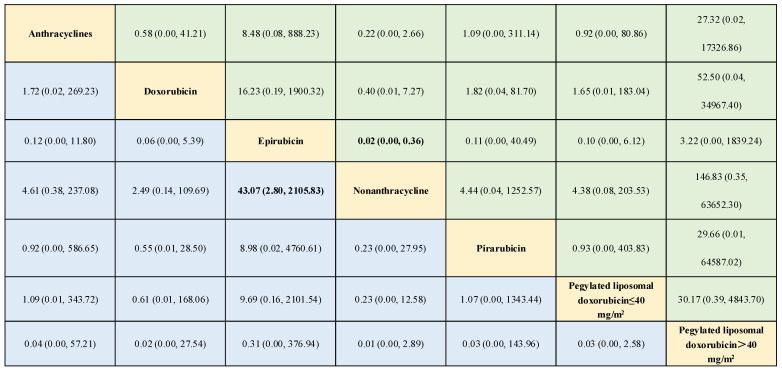
Outcomes of the network meta-analysis of the risk of arrhythmia using the consistency model. The results are the odds ratios (ORs) (95% confidence intervals [CIs]) based on the network meta-analysis of the interventions defined in the columns and rows. Comparisons are shown from left to right. Numbers in bold represent statistically significant results. Different drugs are presented in yellow, and both green and blue represents the ORs of interventions.

**Figure 7 biology-13-00465-f007:**
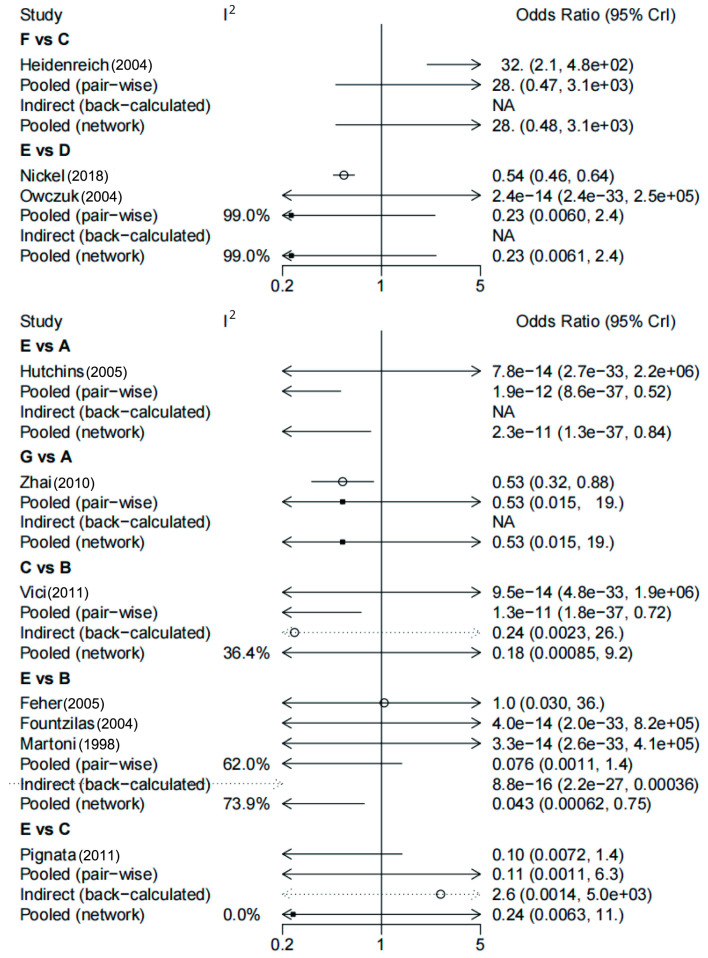
Comparative odds ratios (ORs) for the incidence of arrythmia with different anthracycline treatments and non-anthracycline treatments. The forest plot demonstrates point estimates of the risk ratio surrounded by 95% confidence intervals (CIs) calculated using the random-effects model [25,26,27,28,29,30,31,32,33,34]. A: doxorubicin; B: epirubicin; C: pegylated liposomal doxorubicin ≤ 40 mg/m^2^; D: anthracyclines; E: non-anthracycline antineoplastic drugs; F: pegylated liposomal doxorubicin > 40 mg/m^2^; G: pirarubicin.

**Table 1 biology-13-00465-t001:** Classification and corresponding anthracycline agents.

Classification	Drugs
First-generation anthracycline drugs	Doxorubicin, epirubicin, idarubicin, pirarubicin, valrubicin, aclarubicin, aclacinomycin, daunorubicin, cerubidine, mithracin, plicamycin
Second-generation anthracycline drugs	Anthracyclines, annamycin, sabarubicin, zorubicin
Third-generation anthracycline drugs	Aldoxorubicin, GPX-150, SP1049C, DOXO-EMCH.A
Unclassified anthracycline drugs	Valstar, zoptarelin doxorubicin, luteinizing hormone-releasing hormone, 5-imino-13-deoxydoxorubicin

**Table 2 biology-13-00465-t002:** Characteristics of the included cohort studies and RCTs.

First Author and Year	Study Design	Geographical Location	Funding Sources	Duration of Allocation	Race	Cancer	Duration of Treatment	Duration of Follow-Up	Number (T/C)	Mean Age (Years)		Gender (Male/Female)		Anthracyclines	Treatment	Control	Quality Scores *
										T	C	T	C				
Valagussa 1994 [23]	Cohort study	Italy	Not mentioned	Not mentioned	Not mentioned	Breast cancer	21-day cycle, 4 cycles	8 years	501/324	Not applicable	Not applicable	0/501	0/324	Doxorubicin	Doxorubicin + cyclophosphamide + fluorouracil + methotrexate	Cyclophosphamide + fluorouracil + methotrexate	6
Martoni 1998 [24]	RCT	Italy	Not mentioned	August 1992 to February 1996	Not mentioned	Advanced NSCLC	21-day cycle, 12 cycles	Not mentioned	102/110	62	61	86/16	93/17	Epirubicin	Epirubicin	Vinorelbine	Some concerns
Fargeot 2004 [25]	RCT	French	Pfizer, France	March 1991 to April 2001	Not mentioned	Breast cancer	3 years	72 months	174/164	69	69	0/174	0/164	Epirubicin	Epirubicin + tamoxifen	Tamoxifen	Some concerns
Owczuk 2004 [26]	Cohort study	Poland	Not mentioned	Not mentioned	Not mentioned	Breast cancer	33.1 days	Not mentioned	20/20	52.2	52.8	0/20	0/20	Doxorubicin, epirubicin	Anthracyclines (doxorubicin, epirubicin)	Non-anthracycline antineoplastic drugs	6
Fountzilas 2004 [27]	RCT	Greece	Hellenic Cooperative Oncology Group	January 1999 to April 2002	Not mentioned	Breast cancer	21-day cycle, 6 cycles	23.5 months	162/160	59	59	0/162	0/160	Epirubicin	Epirubicin + paclitaxel	Paclitaxel + carboplatin	Low
Feher 2005 [28]	RCT	Brazil, Czech Republic, Germany, Singapore, USA, Germany	Eli Lilly and Company	Not mentioned	Not mentioned	Breast cancer	28-day cycle, 12 cycles	19.1 months	199/198	68	69	0/199	0/198	Epirubicin	Epirubicin	Gemcitabine	Some concerns
Hutchins 2005 [29]	RCT	USA	National Cancer Institute, Department of Health and Human Services	July 1989 to February 1993	White, non-Hispanic; Black, non-Hispanic; Hispanic; others	Breast cancer	28-day cycle, 6 cycles	2.5 years	669/676	48	48	0/669	0/676	Doxorubicin	Doxorubicin + cyclophosphamide + fluorouracil	Methotrexate + cyclophosphamide + fluorouracil	Some concerns
Pignata 2011 [30]	RCT	Italy	Associazione Italiana per la Ricerca sul Cancro	January 2003 to July 2004	Not mentioned	Ovarian cancer	21-day cycle, 6 cycles	Not mentioned	403/408	57	57	0/403	0/408	Pegylated liposomal doxorubicin	Pegylated liposomal doxorubicin 30 mg/m^2^ + carboplatin	Carboplatin + pegylated liposomal doxorubicin	Low
Nickel 2018 [31]	Cohort study	USA	Not mentioned	January 2010 to December 2015	Not mentioned	Not specified	Not mentioned	6 months	2075/2951	55.3 ± 13.99	59.1 ± 13.8	1099/976	1309/1642	Doxorubicin, epirubicin, daunorubicin	Anthracyclines (doxorubicin, epirubicin, daunorubicin)	Non-anthracycline antineoplastic drugs (bevacizumab, nivolumab, pembrolizumab, ibrutinib, imatinib, ipilimumab, erlotinib, lapatinib, sorafenib, sunitinib, trastuzumab, vemurafenib)	5
Heidenreich 2004 [32]	RCT	Germany	Not mentioned	Not mentioned	Not mentioned	Prostate carcinoma	24 weeks	6.5 months	22/26	69.3 ± 11.5	67.5 ± 11.2	22/0	26/0	Pegylated liposomal doxorubicin	Pegylated liposomal doxorubicin 25 mg/m^2^	Pegylated liposomal doxorubicin 50 mg/m^2^	Some concerns
Zhai 2010 [33]	Cohort study	China	Guangdong National Science	1987–2003	Not mentioned	Non-Hodgkin’s lymphoma	Not mentioned	5 years	205/254	Not applicable	Not applicable	126/79	173/81	Pirarubicin, doxorubicin	Pirarubicin, cyclophosphamide, vincristine, and prednisone	doxorubicin, cyclophosphamide, vincristine, and prednisone	6
Vici 2011 [34]	RCT	Italy	Not mentioned	March 2003 to November 2005	Not mentioned	Breast cancer	21-day cycle, 8 cycles, 28-day cycle, 8 cycles	Not mentioned	54/50	63	61	0/54	0/50	Epirubicin, pegylated liposomal doxorubicin	Epirubicin, vinorelbine	Pegylated liposomal doxorubicin 40 mg/m^2^, vinorelbine	Low

T: treatment arm; C: control arm; RCT: randomized controlled trials. * Cohort studies were assessed by quality assessment according to the Newcastle–Ottawa Quality Assessment Scale. RCTs were assessed according to Risk of Bias-2.

**Table 3 biology-13-00465-t003:** Characteristics of the included single-arm studies.

First Author and Year	Geographical Location	Funding Sources	Duration of Allocation	Cancer	Duration of Follow-Up	Number	Mean Age (Years)	Gender (Male/Female)	Carrier/Tested Subjects	Frequencies%	Type of Arrhythmia	Types of Anthracycline
Abdelsalam 1974 [35]	USA	The National Cancer Institute	Not mentioned	Acute leukemia	Not applicable	61	6.6	43/23	16/61	26.23	(1) (2)	Adriamycin
Daniel 1977 [36]	USA	The Investigational Drug Branch and Cancer Therapy Evaluation Program, Division of Cancer Treatment	Not mentioned	Not applicable	80 days	5613	Not applicable	Not mentioned	45/5613	0.80	(1) (2) (3) (4) (6) (7)	Daunomycin
Bonfante 1982 [37]	Italy	Farmitalia Carlo Erba	Not applicable	Various types of advanced malignancy	Not applicable	108 (100)	50	61/47	24/100	24.00	(1) (2) (7)	4′-epi-doxorubicin
Antman 1984 [38]	USA	Not mentioned	1978–1982	IIB -IVA sarcomas	5 y ears	20	Not mentioned	Not mentioned	1/20	5.00	(2)	Doxorubicin
Jonathan 1987 [39]	USA	Not mentioned	Not mentioned	Not applicable	16 days	29	57 ± 16	16/13	9/29	31.03	(2)	Doxorubicin
Kaukel 1990 [12]	USA	Not mentioned	Not mentioned	Malignant pleural mesothelioma	August 1985 to November 1986	35	68.5	29/6	4/35	11.43	(6) (7)	Pirarubicin
Umsawasdi 1990 [40]	USA	Adria Laboratories, Columbus, Ohio	Not mentioned	Various types of cancer	73 weeks	15	58	10/5	3/15	20.00	(5) (6)	4-demethoxydaunorubicin
Fountzilas 1991 [41]	Greece	Not mentioned	February 1988 to December 1989	Breast cancer	Not mentioned	52 (16)	52	0/52	2/16	12.50	(7)	Epirubicin
Villani 1991 [42]	Italy	Not mentioned	Not mentioned	Gastrointestinal carcinomas, kidney carcinomas, non-small-cell lung carcinomas, unknown primary carcinomas, hepatocellular carcinomas, breast cancer, melanomas, sarcomas, ovarian carcinoma, non-Hodgkin lymphoma	21 days	35	47	24/11	12/35	34.29	(1) (6) (7)	4’-iodo-4’-deoxydoxorubicin
Larsen 1992 [43]	USA	National Institutes of Health Research Career Development	Not mentioned	Acute lymphoblastic leukemia, acute nonlymphocytic leukemia, Hodgkin’s disease, non-Hodgkin’s lymphoma, Wills tumor, neuroblastoma, rhabdomyosarcoma, osteosarcoma, Ewing sarcoma, other sarcomas, teratoma, central nervous system tumors	3 months–21 years	110	15 ± 4	Not mentioned	25/110	22.73	(3) (6) (7)	Anthracyclines
Ando 2000 [44]	Japan	The Ministry of Health and Welfare of Japan	March 1990 and August 1998	Breast cancer	Not applicable	39	46	0/39	11/39	28.21	(1)	Doxorubicin
Gupta 2002 [19]	USA	Not mentioned	Not mentioned	Non-Hodgkin lymphoma, Hodgkin lymphoma, rhabdomyosarcoma, osteogenic sarcoma, acute lymphocytic leukemia, Ewing sarcoma, Wilms tumor, hemangiopericytoma	Not mentioned	64	11 ± 5	34/30	17/64	26.56	(3) (7)	Anthracyclines
Postma 2002 [45]	The Netherlands	Not mentioned	Not applicable	Bone tumor	14.1 years	39	32.5	25/14	12/29	41.38	(1) (2)	Anthracyclines
Sally 2003 [46]	UK	Not mentioned	Not mentioned	Not applicable	Not applicable	14	Not mentioned	13/1	9/14	64.29	(6)	Anthracyclines
Pignata 2006 [30]	Italy	Associazione Italiana per la Ricerca sul Cancro	January 2003 to July 2004	Ovarian cancer	21-day cycle, 6 cycles	50	60	0/50	2/50	4.00	Not applicable	Pegylated liposomal doxorubicin
Chen 2009 [47]	Germany	Not mentioned	2000–2004	Malignant tumor	3.2–8 years	168	8.1 ± 5.3	110/58	4/168	2.38	(6)	Anthracyclines
Ricevuto 2015 [48]	Italy	Not mentioned	Not mentioned	Breast cancer	Not mentioned	24	Not mentioned	0/24	14/24	58.33	(6) (7)	Nonpegylated liposomal doxorubicin
Markman 2017 [49]	USA	Not mentioned	1984–2015	Various types of cancer	14 ± 7 y	134	30.4 ± 5.6	70/64	17/134	12.69	(3) (7)	Anthracyclines

(1): ST-T change; (2): sinus arrhythmia; (3): QT prolongation; (4): bradycardia; (5): atrial fibrillation; (6): tachycardia; (7): others.

**Table 4 biology-13-00465-t004:** Treatment ranked by the probability of the highest risk of arrhythmia.

Medication	Rank Probabilities ^a^						
	Rank 1	Rank 2	Rank 3	Rank 4	Rank 5	Rank 6	Rank 7
Anthracyclines	0.071	0.370	0.274	0.186	0.081	0.013	0.005
Doxorubicin	0.006	0.019	0.015	0.022	0.032	0.304	0.601
Epirubicin	0.008	0.066	0.202	0.412	0.241	0.045	0.026
Non-anthracycline antineoplastic drugs	0.692	0.226	0.070	0.011	0.001	0.000	0.000
Pirarubicin	0.030	0.017	0.019	0.022	0.040	0.593	0.280
Pegylated liposomal doxorubicin ≤40 mg/m^2^	0.171	0.267	0.346	0.172	0.031	0.011	0.001
Pegylated liposomal doxorubicin ˃40 mg/m^2^	0.021	0.035	0.074	0.175	0.573	0.034	0.087

^a^ Rank probabilities were derived from the network meta-analysis. The ranking indicates the probability of being the best treatment, second best treatment, third best treatment, and so on. Rank 7 corresponded to the highest probability of inducing arrhythmia. Rank 1 corresponded to the lowest probability of inducing arrhythmia.

## Data Availability

Data sharing is not applicable to this article as no datasets were generated or analyzed during the current study, but can be check from the supplements.

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
