# Peer review of "Cardiac Arrhythmia Risk after Anti-Cancer Drug Exposure and Related Disease Molecular Imaging Outlook: A Systematic Review, Meta-Analysis, and Network Meta-Analysis"

_biology, 2024, doi:10.3390/biology13070465_

Round 1

Reviewer 1 Report

Comments and Suggestions for Authors

The study entitled “Cardiac Arrhythmia Risk After Anti-cancer Drug Exposure and Related Disease Molecular Imaging Outlook: A Systematic Review, Meta-Analysis, and Network Meta-Analysis” investigates the association between anthracycline chemotherapy and arrhythmia risk. Systematic review and meta-analysis revealed a significant 90% increase in arrhythmia risk and 114% increase in supraventricular arrhythmias with anthracycline use. Epirubicin shows the highest arrhythmia probability. The findings underscore the importance of cardiac monitoring during anthracycline treatment and suggest exploring therapies for associated arrhythmias. Molecular imaging emerges as a crucial tool for understanding drug effects on cardiac electrophysiology. However, before accepting this work, several issues must be addressed, with particular emphasis on the points listed below.

1.      Introduction

a.       The introduction contains several statements that lack precision or proper referencing. For example, stating that chemotherapy is the predominant first-line treatment without specifying the types of cancer or providing evidence to support this claim can be misleading.

b.      The introduction acknowledges that existing evidence on the risk of arrhythmias associated with anthracycline agents is limited and inconclusive, it does not provide  the implications of this uncertainty or discuss potential reasons for conflicting findings in previous studies. Addressing contradictory evidence more directly would demonstrate a thorough understanding of the existing literature and provide a more balanced perspective.

c.       The introduction could smoother transition into the methods section by briefly outlining how the study aims to address the gaps and limitations identified in the existing literature.

d.      The introduction does not clearly define the scope of the review in terms of the types of arrhythmias or patient populations included.

2.      Material and Methods

a.       Although the section mentions a systematic search of various databases, it lacks specificity regarding the search terms used and the process for selecting relevant studies.

b.      Although the section mentions that data extraction was performed by multiple reviewers and discrepancies were resolved through consensus, it does not provide details on how disagreements were resolved or how data quality was ensured.

c.       Some information, such as the description of study quality assessment methods, is repeated multiple times throughout the section, leading to redundancy.

d.      The section mentions assessing the risk of bias due to missing data, but it doesn't specify how missing data in the meta-analysis itself will be handled, such as whether imputation methods will be used or studies with missing data will be excluded.

3.      Results

a.       While the section mentions the number of records identified, screened, and included, it lacks detail on the reasons for exclusion at each stage of the selection process.

b.      Although heterogeneity is mentioned in the meta-analysis, the section does not provide a thorough discussion of its potential sources or implications for the interpretation of results. Addressing heterogeneity and its impact on the findings would improve the robustness of the analysis.

c.       Could you please provide more detail on how the funding sources mentioned in Tables 1 and 2 may have influenced the study outcomes? Specifically, how did industry sponsorship or funding from national science foundations benefit the research methodology or interpretation of results?

d.      Did you perform any sensitivity analyses to assess the robustness of the meta-analysis results, particularly in relation to different study designs or patient populations?

e.       The meta-analysis results indicate a significant association between anthracycline use and an increased risk of arrhythmia. How do these findings contribute to our understanding of the cardiac risks associated with anthracycline therapy?

4.      Discussion

a.       While the author mention the statistically significant increase in the risk of arrhythmia associated with anthracycline use, the discussion lacks in-depth interpretation of the clinical significance of these findings. Can you provide a more thorough analysis of the implications of the observed increase in arrhythmia risk for cancer patients undergoing anthracycline therapy?

b.      The discussion briefly mentions the importance of monitoring ECG changes in patients undergoing anthracycline-based chemotherapy but does not provide practical recommendations for clinical practice. How can the findings of this study be translated into actionable recommendations for oncologists and cardiologists managing cancer patients receiving anthracyclines?

c.       Are there any regulatory implications or recommendations that should be considered based on your findings, such as changes to drug labeling or monitoring protocols for anthracycline use in clinical practice?

Author Response

  1. Introduction
  2. The introduction contains several statements that lack precision or proper referencing. For example, stating that chemotherapy is the predominant first-line treatment without specifying the types of cancer or providing evidence to support this claim can be misleading.

Response: Thank you for your suggestion! We have add more proper and newest references towards this according to your request in line 44-45.

  1. The introduction acknowledges that existing evidence on the risk of arrhythmias associated with anthracycline agents is limited and inconclusive, it does not provide  the implications of this uncertainty or discuss potential reasons for conflicting findings in previous studies. Addressing contradictory evidence more directly would demonstrate a thorough understanding of the existing literature and provide a more balanced perspective.

Response: Thank you for your suggestion! We have add more references and discussed the perspectives according to your request in line 55-62.

  1. The introduction could smoother transition into the methods section by briefly outlining how the study aims to address the gaps and limitations identified in the existing literature.

Response: Thank you for your suggestion! We have edited the description in the introduction part according to your request in line 62-69.

  1. The introduction does not clearly define the scope of the review in terms of the types of arrhythmias or patient populations included.

Response: That is a very good suggestion. The things was that we included all types of carcinoma that applied anthracycline agents. As for the type of arrhythmia, we also took all of them into consideration. Cause we think even though there might be differences in the location of different types of arrhythmias and the type of seizure, if they were caused by anthracycline agnets, they were in need of concern to the clinicians.

  1. Material and Methods
  2. Although the section mentions a systematic search of various databases, it lacks specificity regarding the search terms used and the process for selecting relevant studies.

Response: Actually, we have already provided all the search terms and process in our supplementary materials. We even printscreened every step while we searched (line 89-90).

  1. Although the section mentions that data extraction was performed by multiple reviewers and discrepancies were resolved through consensus, it does not provide details on how disagreements were resolved or how data quality was ensured. 

Response: We have already demonstrated these process between line 106 to 111. Specifically, “The studies were obtained from four medical databases and were independently screened by two reviewers (Hongzheng Li and Wenwen Yang) based on their titles and abstracts. A comprehensive search of the reference lists of the review articles and original studies was performed to identify additional reports. Disagreements were resolved through a consensus or referral to a third reviewer (Xin Zhao) when no consensus was reached. ”

  1. Some information, such as the description of study quality assessment methods, is repeated multiple times throughout the section, leading to redundancy. 

Response: Thank you for your suggestion, we have deleted redundant descriptions towards that.

  1. The section mentions assessing the risk of bias due to missing data, but it doesn't specify how missing data in the meta-analysis itself will be handled, such as whether imputation methods will be used or studies with missing data will be excluded.

Response: It was so nice of your suggestion, we have added these information in this manuscript.

  1. Results
  2. While the section mentions the number of records identified, screened, and included, it lacks detail on the reasons for exclusion at each stage of the selection process.

Response: Detail on the reasons for exclusion is extremely important during meta-analysis, so we did make description about that in previous manuscript (Supplementary Materials: List of articles removed).

  1. Although heterogeneity is mentioned in the meta-analysis, the section does not provide a thorough discussion of its potential sources or implications for the interpretation of results. Addressing heterogeneity and its impact on the findings would improve the robustness of the analysis.

Response: Thank you for your suggestion, we have added some comments in discussion.

  1. Could you please provide more detail on how the funding sources mentioned in Tables 1 and 2 may have influenced the study outcomes? Specifically, how did industry sponsorship or funding from national science foundations benefit the research methodology or interpretation of results?

Response: Thank you for your suggestion, we have added some comments in discussion. Research funded by pharmaceutical companies may be associated with higher bias, but results funded by public health authorities are less likely to produce publication bias than those funded by pharmaceutical companies.

  1. Did you perform any sensitivity analyses to assess the robustness of the meta-analysis results, particularly in relation to different study designs or patient populations?

Response: Sensitivity analysis plays an important role in testing the stability of meta-analysis, so sensitivity analysis should be conducted even if there is no heterogeneity in the selected literature. Therefore, we eliminated the literature one by one and focused on whether there was any influence on the results.

  1. The meta-analysis results indicate a significant association between anthracycline use and an increased risk of arrhythmia. How do these findings contribute to our understanding of the cardiac risks associated with anthracycline therapy?

B in discussion.      The discussion briefly mentions the importance of monitoring ECG changes in patients undergoing anthracycline-based chemotherapy but does not provide practical recommendations for clinical practice. How can the findings of this study be translated into actionable recommendations for oncologists and cardiologists managing cancer patients receiving anthracyclines?

Response to e and b: These are fabulous questions as well as the aim of our research. Our findings show a significant association between anthracyclines use and an increased risk for arrhythmia, especially supraventricular arrhythmia. These results indicated cardiac rhythm should be strictly monitored during the application of anthracyclines in clinical practice, and possible therapy for anthracyclines associated arrhythmia should be explored. We would like to encourage clinicians to monitor ECG in patients treated with anthracyclines. We have already talk about this in abstract as well as discussion.

  1. Discussion
  2. While the author mention the statistically significant increase in the risk of arrhythmia associated with anthracycline use, the discussion lacks in-depth interpretation of the clinical significance of these findings. Can you provide a more thorough analysis of the implications of the observed increase in arrhythmia risk for cancer patients undergoing anthracycline therapy?

Response: Thank you so much for your questions as well as suggestion,we have added some comments in discussion.

  1. Are there any regulatory implications or recommendations that should be considered based on your findings, such as changes to drug labeling or monitoring protocols for anthracycline use in clinical practice?

Response: Thank you so much for your questions as well as suggestion,we have added some comments in discussion.

Reviewer 2 Report

Comments and Suggestions for Authors

Manuscript Title: Cardiac Arrhythmia Risk After Anti-Cancer Drug Exposure and Related Disease Molecular Imaging Outlook: A Systematic Review, Meta-Analysis, and Network Meta-Analysis 

The main interest of the study resides in its description about the association between anthracyclines use and an increased risk for arrhythmia suggesting the importance of molecular imaging technology in studying the mechanism of drug action on cardiac electrophysiology.

Limitation: A larger studies sample size would increase the statistical power and validity of the study. 

Citation numbering should be in accordance with the Journal style.

The information mentioned in Section 2.3, line 86 onwards…. about the pharmacological agents… should be in a table format.

Kindly add a flow chart of patients included in the analysis. 

Do you have data on Anthracycline use, progression of disease, and the risk of arrythmias?

Limitation section should be mentioned separately.

Discussion: The authors should aim for a logical flow of the text that enhances readability and navigation.

Comments on the Quality of English Language

Minor editing needed.

Author Response

Manuscript Title: Cardiac Arrhythmia Risk After Anti-Cancer Drug Exposure and Related Disease Molecular Imaging Outlook: A Systematic Review, Meta-Analysis, and Network Meta-Analysis 

Response: The main interest of the study resides in its description about the association between anthracyclines use and an increased risk for arrhythmia suggesting the importance of molecular imaging technology in studying the mechanism of drug action on cardiac electrophysiology.

Limitation: A larger studies sample size would increase the statistical power and validity of the study. 

Response: Thank you so much for your suggestion, large sample size is extremely necessary, and we even believe that we can use Mimic II, UK biobank, and medical insurance database to explore that. What we can do now is to sort through the published studies, and we think that is one of the accessible ways to answer that question. In the future, we would like to see the registration studies for anthracyclines, which are necessary for the safe use of them.

Citation numbering should be in accordance with the Journal style.

Response: Thank you so much for your suggestion, we have edited the citation format (line 464-632).

The information mentioned in Section 2.3, line 86 onwards…. about the pharmacological agents… should be in a table format.

Response: Thank you so much for your suggestion, we have added a table concerning to these agents (Table 1).

Kindly add a flow chart of patients included in the analysis. 

Response: Thank you for your suggestion, please review the flowchart below with the number of patients to better understand the patients involved in the analysis. We have added according to your request in line 192, thanks again for the suggestion.

Do you have data on Anthracycline use, progression of disease, and the risk of arrythmias?

Response: Based on references [1-2], we can see that anthracycline agents are widely used in the treatment of carcinoma, but data on their use, disease progression, and risk of arrhythmias are indeed a complex and critical topic. First, the use of anthracycline agents varies between disease stages and patient populations, so the evaluation of their effects requires a combination of factors. Regarding disease progression, anthracyclines are able to control the disease to some extent, but they are not effective in all patients. Some studies suggest that long-term use of anthracycline agents may delay disease progression, but resistance or adverse effects may also occur. Therefore, while using anthracycline agents, it is necessary to pay close attention to the changes in the patient's condition, especially on ECG changes and adjust the treatment regimen in time. As for the risk of arrhythmias, anthracycline agents do have potential harms in this regard. Some patients may experience adverse effects such as arrhythmias after the use of cyclic drugs, which may be related to the dosage of the drug, the duration of use, and individual differences in patients, and the incidence of these studies has been as high as ST-T changes, QT interval prolongation, and atrial fibrillation[11-15].

Limitation section should be mentioned separately.

Response: We are grateful for your valuable advice. The restrictions have been revised and a separate paragraph has been deliberately arranged to elaborate on them to ensure accuracy and clarity of the content in line 377-394.

Discussion: The authors should aim for a logical flow of the text that enhances readability and navigation.

Response: We greatly appreciate your valuable comments and suggestions, which are crucial for enhancing the quality of our manuscript.We strongly agree with your suggestion about the logical flow of text. Indeed, logical fluency is an indispensable and important element of a good essay. Not only does it enhance the readability of the article and make it easier for readers to follow the author's train of thought, but it also improves the navigation of the article so that readers can find what they're interested in faster. When revising, we pay more attention to the logical structure of the text to ensure that the connection between paragraphs is natural and smooth. At the same time, we will also pay attention to the use of appropriate inflections and conjunctions to enhance the coherence and logic of the sentence. In addition, we will also strengthen the proofreading and revision of the article to ensure the accuracy and standardization of the text. We believe that through continuous efforts, the quality of our manuscripts will be further improved and bring a better reading experience to readers. Thank you again for your valuable advice, and we look forward to continuing to work with you to improve the quality and impact of your articles.

 Comments on the Quality of English Language

Minor editing needed.

Response: Thank you for your suggestions. We have edited the manuscript by native British according to your suggestions and corrected the spelling mistakes.

Round 2

Reviewer 1 Report

Comments and Suggestions for Authors

The authors have addressed all my queries.

Author Response

Comments1:The authors have addressed all my queries.

Reply to comments 1: Thank you very much for your recognition! 

Comments 2:New Table 1, which was created after Reviewer 1's recommendation, needs more organization. A horizontal line should be used to delimit each row in the table, or each row could have a background of a different color. These comments should be applied to Table 2 and Table 3.

Reply: Dear editor, thank you so much for your comments, in this manuscript we have edited all the tables according to Template, we hope that these modifications will meet your requirements.

Best wishes to you all.
